# How Plant Toxins Cause Early Larval Mortality in Herbivorous Insects: An Explanation by Modeling the Net Energy Curve

**DOI:** 10.3390/toxins16020072

**Published:** 2024-02-01

**Authors:** Suman Chakraborty, Stefan Schuster

**Affiliations:** 1Department of Bioinformatics, Matthias Schleiden Institute, Friedrich Schiller University Jena, Ernst-Abbe-Pl. 2, 07743 Jena, Germany; suman.chakraborty@uni-jena.de; 2International Max Planck Research School “Chemical Communication in Ecological Systems”, 07745 Jena, Germany

**Keywords:** glucosinolates (GLSs), isothiocyanates (ITCs), larval death, herbivory, net-energy, metabolic cost, non-autonomous differential equations, optimal foraging theory

## Abstract

Plants store chemical defenses that act as toxins against herbivores, such as toxic isothiocyanates (ITCs) in Brassica plants, hydrolyzed from glucosinolate (GLS) precursors. The fitness of herbivorous larvae can be strongly affected by these toxins, causing immature death. We modeled this phenomenon using a set of ordinary differential equations and established a direct relationship between feeding, toxin exposure, and the net energy of a larva, where the fitness of an organism is proportional to its net energy according to optimal foraging theory. Optimal foraging theory is widely used in ecology to model the feeding and searching behavior of organisms. Although feeding provides energy gain, plant toxins and foraging cause energy loss for the larvae. Our equations explain that toxin exposure and foraging can sharply reduce larval net energy to zero at an instar. Since herbivory needs energy, the only choice left for a larva is to stop feeding at that time point. If that is significantly earlier than the end of the last instar stage, the larva dies without food. Thus, we show that plant toxins can cause immature death in larvae from the perspective of optimal foraging theory.

## 1. Introduction

Plants of the Brassicaceae family store glucosinolates (GLSs) and their corresponding activating enzyme (myrosinase) in their leaves to defend themselves against herbivore attacks [1,2,3]. The plant stores GLSs and myrosinase in separate leaf compartments to protect itself. Upon herbivory, the components are brought together, which initiates GLS hydrolysis via myrosinase [2,3] to produce isothiocyanates (ITCs), nitriles, or epithionitriles [4]. These hydrolyzed products act as toxins, growth inhibitors, and deterrents to leaf-chewing lepidopteran larvae [4]. Among all those chemicals, ITCs are the most toxic, immensely damaging larval fitness [5,6].

In response to plant defenses, lepidopteran insects have evolved special adaptations [7], called counter-defenses. Insects with preemptive counter-defenses prevent (not perfectly) the formation of ITCs to avoid toxin exposure [8,9,10,11]. For example, some GLS-metabolizing insects, such as *Plutella xylostella*, desulfate GLSs before hydrolysis [9], and *Pieris rapae* can redirect GLS hydrolysis to produce nitriles [11]. Some GLS-sequestering specialists, such as *Athalia rosae* L. and *Phyllotreta armoraciae*, rapidly absorb GLSs before hydrolysis [8,10,12].

Insects with direct counter-defenses, on the other hand, do not have any mechanisms to circumvent ITC formation. However, they can metabolize the hydrolyzed ITCs to nontoxic conjugates of the tripeptide L-glutathione (GSH) [7,13,14]. Generalists, for example, *Spodoptera littoralis, Spodoptera exigua, Trichoplusia ni, Mamestra brassicae*, and *Helicoverpa armigera*, usually employ this type of ITC-detoxification strategy. Experimental results suggest that direct counter-defense is less efficient, meaning that the concentration of unmetabolized or free ITC is comparatively higher with direct counter-defense than for pre-emptory detoxification system [13,14]. This is substantiated by mathematical modeling [15].

One of the major negative effects of ITCs on feeding larvae is a high mortality rate. For example, both *M. brassicae* and *P. rapae* larvae show low survival rates on Brassicaceae plants [16]; an average mortality of 80% was reported in generalist cabbage moth (*Mamestra brassicae*) for larvae of the first instar while feeding on white cabbage (*Brassicae oleracea var. capitata*) [17]. ITCs are very effective in increasing mortality in specialist small white *Pieris rapae* larvae [18]. *Mamestra brassicae* larvae have low chances of survival against the aliphatic GLSs (sinigrin and gluconapin) contained in *Brassica oleracea* leaves [19]. Allyl isothiocyanate (AITC) can raise mortality up to 100% in larvae of masked chafer beetles, *Cyclocephala* spp. *Latreille* [20,21], *Sitophilus zeamais*, *Rhyzopertha dominica*, and *Tribolium castaneum* [22].

The fast decaying fitness of a larva does not only lead to high mortality but also to other adverse effects, such as reduced growth, slow feeding, delayed development, etc. [5,13]. Therefore, it is important to determine how the fitness of a larva is correlated to the interplay between nutritional gain and the various types of toxin (ITCs) costs, owing to herbivory. However, in our study, we only focused on how toxin cost leads to larval mortality. In optimal foraging theory (OFT) [23,24], a long-known concept (originally established for predation) is that species increase their fitness by maximizing the net rate of energy intake from feeding [25,26,27,28]. The net energy benefit of an organism comes from feeding, but foraging causes net energy loss [27,28]. Moreover, herbivory stops if the net energy benefit is zero [27].

Foraging costs should be common for every species if food resources are not abundant [29,30]. However, in the case of toxic-plant-feeding larvae, toxin exposure is the major cost, because fitness is negatively correlated to toxin (ITCs) concentration in the larval gut [5,13]. Therefore, given the aspects of the optimal foraging models under consideration, we accommodated ITC exposure as an additional cost to a leaf-chewing larva on a Brassicaceae plant. Through herbivory, the net energy of a larva is gained from nutrition but lost through the metabolic cost of ITCs, together with foraging, as shown in the schematic diagram in Figure 1.

We formulated the larval net energy curve by developing a system of non-autonomous ordinary differential equations (ODEs), because ODEs are widely used in mathematical biology to describe dynamic processes [27,28,31]. From the behavior of the net energy curve, we explained the relationship between net energy and larval mortality. Our model equations establish a continuous relationship between herbivory, GLS ingestion, exposure to ITCs, and net energy of a larva as its instar stage progresses. From that relationship, we obtained the exact time at which the larval net energy is zero. Theoretically, it is the time point at which a larva stops feeding, because further feeding leads to negative net energy. If the net energy is zero long before the final instar, the larva dies due to lack of nutrition. Thus, from the perspective of optimality theory [32], we explain how decaying fitness causes mortality in the larval stages of insect herbivores on toxic Brassicaceae host plants.

## 2. Results

We summarize our main results as follows:Herbivory per larva is a strictly monotonic increasing quadratic function, as shown in Equation (Equation 2) and Figure 2A, which is a good fit to previously published data [33].The GLSs ingested by a larva are proportional to herbivory, as shown in Equation (Equation 4) and Figure 2A.The concentration of ITCs in the larval gut is a strictly monotonic increasing function, as shown in Equation (Equation 7) and Figure 2B.Due to the adverse effect of ITCs (plant toxins), the larval net energy becomes zero at a certain time point (*T*). This may happen in any instar, as shown in Equation (Equation 10) and Figure 3 and Figure 4.When net energy becomes zero, a larva stops herbivory because further feeding leads to negative net energy, as shown in Equations (Equation 11) and (Equation 15) and Figure 3. The energy benefit of herbivory and the metabolic costs of toxin exposure and foraging determine the particular time, *T*, as shown in Figure 4.The time point, *T*, when herbivory stops can be obtained numerically from Equation (Equation 11). Analytically, *T* can be calculated if some approximations are used, as shown in Equation (Equation 15).If the net energy reaches zero at an early time point (Figure 4), i.e., *T* is small in comparison to larval development time, the larva dies prematurely. A larva does not necessarily die as soon as it stops herbivory; there could be a time lag, but eventually it dies without nutrition. If net energy becomes zero close to the end of the larval instar stages (Figure 4), i.e., *T* is approximately equal to larval development time, then the larva survives the plant toxin.

## 3. Discussion

We developed non-autonomous ordinary differential equations (ODEs) under the framework of optimal foraging theory (OFT) to obtain results that help explain the early larval mortality caused by plant toxins, as shown in Equations (Equation 11) and (Equation 15). Our use of non-autonomous equations was motivated by the reasoning that larval development is driven by a genetic program that is implemented in time. All equations can be solved analytically. A major result is that net energy intake can become zero during any instar, depending on the activity and concentration of the toxin. This can cause the immature death of the larva.

Optimal foraging theory (OFT) has been applied to determine the foraging behavior of the European starling [34], honeybee workers [35], centrarchids (white crappie and bluegill) [36], muskoxen [37], and so on. However, applying OFT to field studies can be challenging [38]. Especially, field conditions may introduce complexities that are not considered in theoretical models [39]. Regarding our model of larval feeding behavior and exposure to plant toxins, certain observations can be made in the field. For example, the mean value expressing the maximum time of larval herbivory (*T*) can be measured; parameters related to larval herbivory, such as θ,κ, can be estimated from larval leaf consumption data [33]; parameters related to toxin exposure, such as β,γ, can be estimated by fitting our ITC curve to data [13]. Moreover, a relationship between the costs of toxins and foraging (μ and δ, respectively) can be established from the analytical solution (Equation 15).

Here, we adopted a deterministic approach. Deterministic models are widely used in various subfields of mathematical biology, such as epidemiology [40,41], population dynamics [42,43,44,45], enzyme kinetics [15,31,46,47,48], optimal foraging [27,28], and so on. In parallel to deterministic techniques, stochastic methods are also extensively applied to explain biological phenomena. For example, geometric Brownian motion (GBM) is a stochastic process in which the logarithm of the variable follows Brownian motion with a drift. It can be described using a particular stochastic differential equation (SDE) [49,50,51,52]. GBM can be used to model the population dynamics of bacteria [53,54], for example, under shock treatment with toxins or antibiotics killing a large part of the population [55]. In mathematical language, this reset means an abrupt reduction in the process variable [52,55]. Similarly, we mathematically explained that larval net energy can be abruptly reduced to zero in an immature stage of development, as shown in Equations (Equation 11) and (Equation 15). However, while bacteria can start dividing again and, thus, their population can recover, insect larvae show different growth dynamics in that they “age” and cannot recover as easily as bacteria. We found that our deterministic approach, which is procedurally different from the above-mentioned stochastic methods, is sufficient to describe the phenomenon under study.

Generalist insect herbivores are usually highly affected because their toxin (*I*) exposure is high due to their inefficient counter-defense mechanisms [13,14,15]. For example, the amount of ITCs is extremely high in the feces of generalist Lepidoptera (*S. littoralis, H. armigera, M. brassicae*, and *T. ni*) in comparison to that in specialist Lepidoptera (*P. xylostella* and *P. rapae*) of Brassicaceae host plants [13,14]. Therefore, the toxin cost function (μI) increases faster for generalists, which causes a fast-degrading net energy curve in Equation (Equation 10). This explains why immature deaths (high larval mortality) are highly common among generalist insects [16,17,19].

High mortality can sometimes also be seen in larvae that are specialists for Brassicaceae [16,18,20]. For example, although *P. rapae* is a specialist against the defenses of Brassica plants, ITCs can cause immature death in *P. rapae* larvae [16,18]. This phenomenon can be explained using Equation (Equation 10). Although, toxin (ITCs) exposure is comparatively low for specialist larvae (both GLS-metabolizing and -sequestering), it is not negligible [10,13,15,56]. Earlier, we substantiated, using a model, that despite having efficient counter-defense systems, specialist larvae can be affected by plant toxins [15]. Since there is an exposure to toxins with an associated cost μ (a proportionality constant), Equation (Equation 8), larval net energy can be decreased by ITCs (Figure 3 and Figure 4). Therefore, the net energy of a specialist larva can be zero early on (before its last instar ends), depending on the cost (μ) associated with the toxins (*I*). So, we can say that specialists should be less affected by plant defenses than generalists due to their low toxin exposure, but they are not completely immune to plant defenses [15,57].

Generally, in experimental studies, the fitness of a larva is determined using the time courses of larval body size and weight [13,58,59,60,61,62,63,64]. The data suggest that a larva develops exponentially (or sometimes linearly) until its last instar [58,63,64]. In addition to early larval mortality, plant toxins slow the development curve (i.e., slow weight and size gain) of a larva [4,65,66,67]. Larval growth may be explained by its net energy function. At the start, net energy intake is very low, so that growth is slow. Later, the growth rate increases because net energy is increased. In the end, net energy intake becomes very low again, reaching zero at the point of stopping herbivory, which indicates that the larval growth curve should also decrease [63,64]. In future modeling studies, it would be worth elucidating the relationship between larval development (mass and size increases) and the net energy reserve of a larva in more detail.

We can generalize our results to other plant families. High mortality in herbivorous insects when faced with plant toxins is a widely common phenomenon [68]. For example, the ingestion of luteolin flavone caused 43% mortality in *S. exigua* caterpillars [69]. Codling moth (*Cydia pomonella*) caterpillars have a cumulative mortality of 23% in the first instar, 33.6% in the second instar, 75.2% in the third instar, and 96.8% in the fourth instar [33] on apple plant leaves, possibly due to the phlorizin defense. The crude extract of a herbaceous plant, *Cynodon dactylon*, caused 75% mortality in *Spodoptera litura* larvae [70]. High mortality in *Culex quinquefasciatus* mosquitoes was found when facing the chemical defenses of five weed plants (*Convulvulus arvensis, Chenopodium murale, Tribulus terrestris, Trianthema portulacastrum*, and *Achyranthes aspera*) [71]. Therefore, we think that, with small modifications, our model can be applied to explain the generally high larval mortality owing to plant toxins.

Our model includes only the direct role of plant toxins [72,73]. However, plant defenses also cause high mortality in insect herbivores by attracting the natural enemies (predator and parasitoid) [42,74,75,76]. This is called the indirect effect of plant defenses [72]. For example, *Cotesia rubecula* wasps are attracted by nitriles [77], and *Trichogramma chilonis* wasps are recruited by isothiocyanates in Brassica plants [78]. Predation and parasitism by natural enemies, causing immature death in larvae, is substantiated by mathematical modeling in population ecology [42,79,80]. However, we did not incorporate this indirect role of plant defenses, rather focusing only on the direct, adverse effect of plant toxins on the fitness of a larva.

Host plant toxins only kill a certain percentage of larvae during herbivory [33,69,70]. This causes a major problem from crop protection against pest infestations [81]. Several approaches are implemented to stop insect pests, one of which is the application of insecticides [82]. Insecticides are treated as novel compounds that are toxic to the feeding larvae [83]. Since insects are usually not adapted to novel toxins, larval fitness can be rapidly reduced to death [71,84]. Therefore, our model can be applied to explain the early mortality of larvae caused by insecticide application. However, insecticides can have negative effects on our environment as well as cause serious public health issues [84,85]. Moreover, insecticide resistance in insect pests raises critical concerns [86], wihch is why alternative safe approaches, such as intercropping, sterile insect techniques, etc., are practiced for crop protection [87,88].

## 4. Materials and Methods

Lepidepteran larvae grow in instars. Instar-specific increases in larval mass and herbivory have been reported [33,58,63,64]. Therefore, we relied on the leaf consumption data from the codling moth (*Cydia pomonella*) to express herbivory as a function of instar or time. Published data [33] regarding herbivory by *C. pomonella* on apple leaves suggest that the cumulative leaf consumption (i.e., herbivory) per larva with respect to its instar follows a quadratic or parabolic curve. The mean data points [33] are shown in Table 1 and plotted in Figure 5. Since *C. pomonella* belongs to the Lepidopteran order, a similar herbivory function can be assumed for other species of that order due to the similarities in biology. Moreover, by way of example, we focused, in this study, on Brassicaceae plants. However, the described phenomena are very general for the feeding of insect larvae on plants (see Section 3).

For simplicity, if we assume that the herbivory growth function is differentiable even at the transitions between instars, then the data points (Figure 5) can be modelled deterministically using a non-autonomous ordinary differential equation. Let H(t) be the amount of herbivory of a larva at time *t*, where larval instar continuously increases with its development time. Herbivory by a larva increases as its instar progress or the larva develops [33], where larval development time is also its herbivory time. Therefore, we denote the herbivory growth rate per unit time (for a larva) using a constant θ, i.e., the herbivory growth rate is proportional to development or herbivory time with the proportionality constant θ. The rate (growth) equation of larval herbivory is then:(1)$$\frac{1}{t}\;\frac{dH}{dt}=\theta$$

Here and below, we use non-autonomous differential equations (i.e., equations explicitly depending on time) because we consider larval development as driven by a genetic program that is implemented over time. Since there is no herbivory until the larva is hatched from an egg, the initial condition is H(t=0)=0. Solving Equation (Equation 1) with this initial condition, we obtain:(2)$$H=\frac{\theta t^{2}}{2}$$

Thus, herbivory per larva in Equation (Equation 2) is a quadratic or parabolic function of time, as shown in Figure 2A, which matches the data points [33] in Figure 5. More exactly, the parameter θ depends on the instar. For simplicity, we neglect this effect and consider θ to be constant throughout. Note that when larval development ends, herbivory stops. Mathematically, this is defined as:(3)$$\;\mathrm{Herbivory}=\left\{\begin{array}{c} 0,\;\mathrm{at}\;t=0\\ H(t),\;\mathrm{at}\;0<t\leq b,\\ 0,\;\mathrm{at}\;t>b \end{array}\right.$$
where *b* is the end time point of larval development.

Since GLSs are entirely contained in the plant leaves, and our model does not take into account spatial heterogeneity, it is plausible to assume that the ingestion of GLSs by a leaf-chewing larva is proportional to its herbivor. Let G(t) be the GLSs ingested by a larva at time t∈[0,b]; the GLS ingestion is:(4)$$G=\frac{\alpha t^{2}}{2},\;\;\mathrm{where}\;\mathrm{initially}\;G(t=0)=0$$
where α is the GLS ingestion growth rate constant per unit time *t*; i.e., the growth rate of GLS ingestion ($\frac{dG}{dt}$) is defined as αt, which linearly increases with time *t*. The quadratic or parabolic curve of GLS ingestion, in Equation (Equation 4), is shown in Figure 2A.

The ingested plant GLSs are degraded to form ITCs, other less toxic products [4], and nontoxic products (for pre-emptive counter-defense only) [9,11,15]. Moreover, GLSs can be sequestered by sequestering specialists [8,10], which can also be included in the degradation of ingested plant GLSs. Let η be the rate at which the ingested GLSs are degraded, and let F(t) be the free GLS content in the larval gut after degradation at time t∈[0,b]. Therefore, the growth rate of the free GLS content in larval gut is:(5)$$\frac{dF}{dt}=\alpha t-\eta F$$

Experimental studies found no traces of free GLSs (*F*) in the larval gut [5,13,14]. For example, *Spodoptera littoralis* and *Mamestra brassicae* larvae were fed on *Arabidopsis thaliana* leaves, and feces of those larvae were collected daily. However, free GLSs were not detected in the larval feces [13]. Therefore, we assume that *F* stays in a quasi steady state [47,89] close to 0 on each day during the entire larval development period. In particular, this implies that the net increase in the free GLS concentration (after ingestion) in the larval gut is negligibly small compared to degradation, i.e. $\frac{dF}{dt}\approx 0$ in Equation (Equation 5). These assumptions prove that plant GLS ingestion and degradation are simultaneous processes, explained by the following calculations:$$\begin{array}{cc} \frac{dF}{dt} & \approx 0\\ \; & \Rightarrow \;\eta F\approx \alpha t\\ \; & \Rightarrow \frac{\alpha t}{\eta }\approx 0\;\;\forall \;\;t>0,\;\mathrm{since}\;F\approx 0\;\mathrm{is}\;\mathrm{the}\;\mathrm{quasi}\;\mathrm{steady}\;\mathrm{state}\\ \; & \Rightarrow \alpha \ll \eta \end{array}$$

This implies that ingested GLSs are degraded immediately. Moreover, ηF≈αt means that GLSs’ degradation to ITCs and other products in the larval gut is proportional to time *t*.

Since ITCs are the most toxic (having direct fitness cost for a larva) among all the products, we modeled only the dynamic ITC concentration in the larval gut. Let β be the hydrolysis rate at which ITCs are formed from GLSs, and let γ be the excretion rate and rate of nontoxic conjugate product formation (only for direct counter-defense) [7,13,14,15]. Assuming I(t) is the concentration of ITCs in the larval gut at time t∈[0,b], the rate equation for ITC exposure is:(6)$$\frac{dI}{dt}=\beta t-\gamma I$$

The initial condition for the produced ITCs is I(t=0)=0. Since Equation (Equation 6) is a first-order linear ODE system, we can analytically solve it using the integrating factor method (see Section A.1):(7)$$I=\frac{\beta }{\gamma^{2}}\left(\gamma t-1+e^{-\gamma t}\right)$$
where Equation (Equation 7) is a strictly monotonic increasing function of time (for proof, see Section A.3). The graphical presentation of *I* for definite parameter values is shown in Figure 2B.

The net energy curve for a larva can be determined from the difference between the energy-benefit function of herbivory and metabolic cost functions of toxin and foraging, as shown in Figure 1. Below, we develop these functions:

**Energy benefit:** A larva obtains energy benefit from herbivory. Therefore, this function is proportional to herbivory: Equations (Equation 1) and (Equation 2). Let the increase in energy growth rate per unit time due to the growth of larvae be a constant κ. Then, the energy growth rate is a linear function κt (like Equation (Equation 1)) and larval energy is a monotonic increasing quadratic function of time $\frac{\kappa t^{2}}{2}$ (like Equation (Equation 2)) if metabolic costs (of toxin and foraging) are absent.**Toxin cost:** The metabolic costs of toxins in a larva should be proportional to the ITC concentration *I*. Assuming μ is the proportionality constant, a simple function for the metabolic cost of toxin is μI.**Foraging cost:** This is the metabolic cost of herbivory, i.e., the energy expended for herbivory. Therefore, the foraging cost is proportional to the herbivory function in Equation (Equation 2), denoted as $\delta t^{2}$, where δ is the proportionality constant.

By denoting the net energy of a larva at time t∈[0,b] as $E_{N}$, the change in net energy is:(8)$$\frac{dE_{N}}{dt}=\kappa t-\mu I-\delta t^{2}$$

Using the value of *I* from Equation (Equation 7), Equation (Equation 8) can be rewritten as:(9)$$\frac{dE_{N}}{dt}=\kappa t-\frac{\mu \beta }{\gamma^{2}}\left(\gamma t-1+e^{-\gamma t}\right)-\delta t^{2}$$

A larva needs basic energy (some positive value) to commence herbivory. This could be called the innate energy, already present in a larva just after it is hatched from the egg. So, initially, net energy is equal to the innate energy, i.e., $E_{N}\left(t=0\right)=E_{N_{0}}>0$. Solving Equation (Equation 8) with this initial condition, we obtain (see Section A.2):(10)$$E_{N}=\frac{\mu \beta t}{\gamma^{2}}-\left(\frac{\mu \beta }{\gamma }-\kappa \right)\frac{t^{2}}{2}+\frac{\mu \beta e^{-\gamma t}}{\gamma^{3}}-\frac{\delta t^{3}}{3}+\left(E_{N_{0}}-\frac{\mu \beta }{\gamma^{3}}\right)$$

The curve corresponding to Equation (Equation 10) for larval net energy ($E_{N}$) is monotonically increasing first, but monotonically decreasing later, as shown in Figure 3. The proof of the curve behavior is given in Section A.3.

Moreover, $E_{N}\left(t\right)$ becomes zero at a certain time point. Let us denote that intersection point of $E_{N}$ and the time axis (*t*) as *T*:(11)$$E_{N}\left(T\right)=\frac{\mu \beta T}{\gamma^{2}}-\left(\frac{\mu \beta }{\gamma }-\kappa \right)\frac{T^{2}}{2}+\frac{\mu \beta e^{-\gamma T}}{\gamma^{3}}-\frac{\delta T^{3}}{3}+\left(E_{N_{0}}-\frac{\mu \beta }{\gamma^{3}}\right)=0$$

*T* represents the maximum time of herbivory for a larva. If *T* is less than the larval developmental time (*b*), the larva does not survive. Since the net energy is zero at time *T*, the larva is forced to stop herbivory. Otherwise, larval net energy becomes negative via herbivory, which is impossible, as shown in Figure 3.

**Remark** **1.**
*The stopping time of herbivory (T) can be quite short or long (close to the end of larval development time), as shown in Figure 4. The net energy curve depends on the values of κ,β,γ,δ, and μ. Different groups of insect larvae should have different net energy curves (Figure 4), according to their adaptiveness to plant toxins.*


Although the net energy function (Figure 3) guarantees the existence of the time point *T*, there is no analytical way to explicitly calculate *T*. Numerical methods (such as bisection, Newton–Raphson, secant, false position, etc.) can be used to find *T*. Moreover, proposing an analytical approximation to obtain *T* is of interest, as we describe in the following subsection.

### Analytical Approximate Calculation

Since an exponential function progresses faster than a polynomial function, at high values of *t*, we can ignore the negative exponential term $e^{-\gamma t}$ in Equation (Equation 9). Thus, Equation (Equation 9) is simplified to:(12)$$E_{N}\approx \frac{\mu \beta t}{\gamma^{2}}-\left(\frac{\mu \beta }{\gamma }-\kappa \right)\frac{t^{2}}{2}-\frac{\delta t^{3}}{3}+\left(E_{N_{0}}-\frac{\mu \beta }{\gamma^{3}}\right)$$
where $E_{N_{0}}-\frac{\mu \beta }{\gamma^{3}}\geq 0$, otherwise $E_{N}$ becomes negative at t=0, which is impossible. However, we make a further approximation by assuming that larval net energy is negligible at t=0, i.e., $E_{N_{0}}-\frac{\mu \beta }{\gamma^{3}}\approx 0$. Thus, Equation (Equation 12) can be written as:(13)$$E_{N}\approx \frac{\mu \beta t}{\gamma^{2}}-\left(\frac{\mu \beta }{\gamma }-\kappa \right)\frac{t^{2}}{2}-\frac{\delta t^{3}}{3}$$

Equating Equation (Equation 13) to zero at t=T, we get:(14)$$\frac{\mu \beta }{\gamma^{2}}-\left(\frac{\mu \beta }{\gamma }-\kappa \right)\frac{T}{2}-\frac{\delta T^{2}}{3}=0,\;\mathrm{since}\;T\neq 0$$

Solving Equation (Equation 14), we obtain:(15)$$T=\frac{3\left(\kappa -\frac{\mu \beta }{\gamma }\right)+\sqrt{9{\left(\frac{\mu \beta }{\gamma }-\kappa \right)}^{2}+\frac{24\mu \beta \delta }{\gamma^{2}}}}{2\delta },\;(\mathrm{without}\;\mathrm{the}\;\mathrm{negative}\;\mathrm{solution})$$
where *T* is positive if we consider only the positive square root, which always produces a value greater than $\left|3\left(\kappa -\frac{\mu \beta }{\gamma }\right)\right|$. Interestingly, *T* remains positive irrespective of whether κ≥ or $<\frac{\mu \beta }{\gamma }$. If the parameters (κ,β,γ,δ,μ) are known or estimated, the approximate time when a larva stops herbivory can be obtained from Equation (Equation 15).

## Figures and Tables

**Figure 1 toxins-16-00072-f001:**
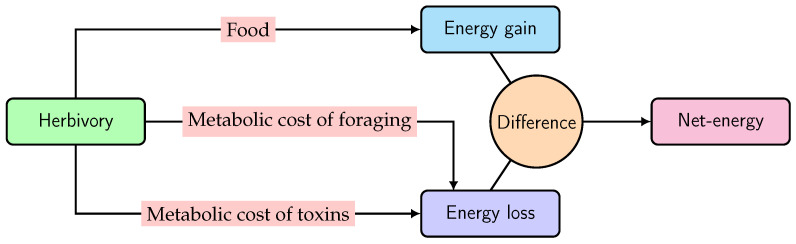
Scheme for the net energy of a larva, obtained through herbivory.

**Figure 2 toxins-16-00072-f002:**
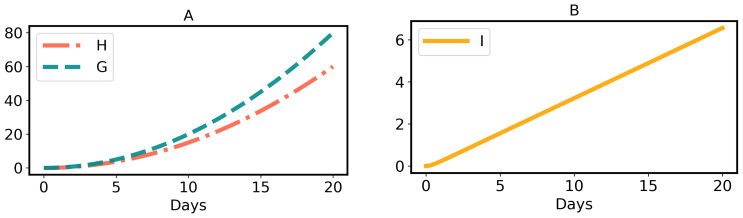
(**A**) Herbivory (*H*) and GLS ingestion (*G*) by a larva, as calculated using Equations (Equation 2) and (Equation 4), respectively. Parameters: θ=0.3, α=0.4. (**B**) ITC (I) exposure of a larva, as calculated using Equation (Equation 7). Parameters: β=0.2, γ=0.8.

**Figure 3 toxins-16-00072-f003:**
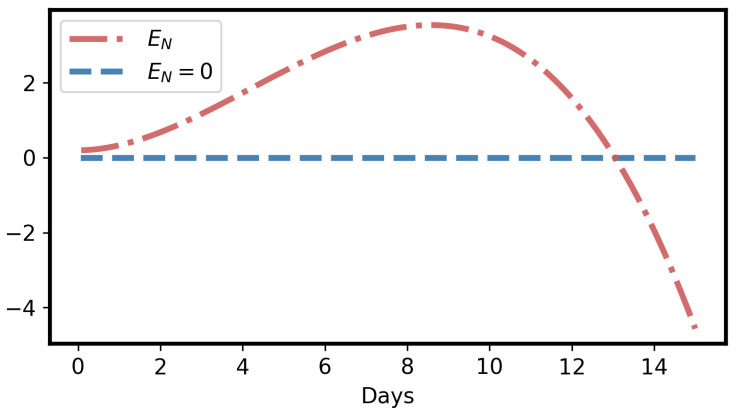
Net energy ($E_{N}$) of a larva becomes 0 at some time point, initial value $E_{N_{0}}=0.2$; parameters κ=0.3, μ=0.1, δ=0.03, β, γ are the same as in Figure 2.

**Figure 4 toxins-16-00072-f004:**
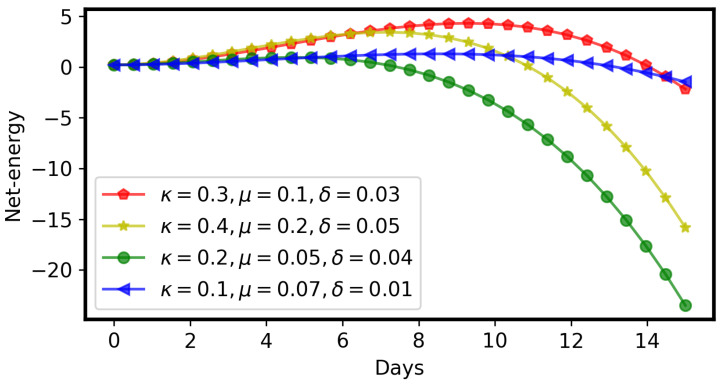
Net energy ($E_{N}$) of a larva can be zero early or later, depending upon the parameters of benefits (κ) and costs (μ and δ).

**Figure 5 toxins-16-00072-f005:**
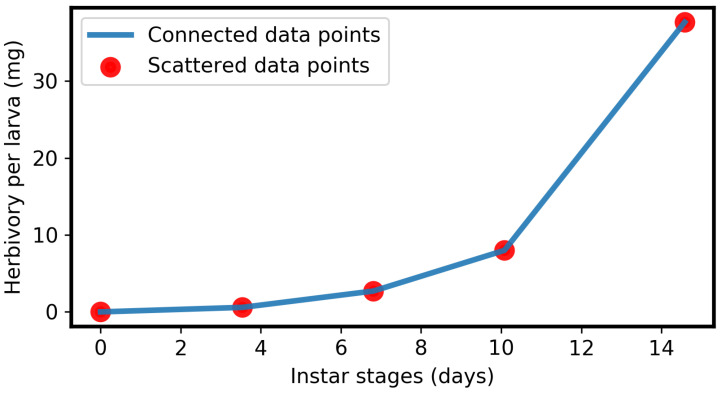
Plot of data points from Table 1.

**Table 1 toxins-16-00072-t001:** Mean data of cumulative leaf consumption (or herbivory) per larva for different instars of codling moth (*Cydia pomonella*), originally published in [33].

Instar	Cumulative Leaf Consumption (or Herbivory) per Larva (Mean)	Duration of Instar (Mean)
1	0.588 mg	3.54 days
2	2.713 mg	3.27 days
3	8.001 mg	3.27 days
4	37.667 mg	4.5 days

## Data Availability

Data are contained within the article.

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
