# Peer review of "How Plant Toxins Cause Early Larval Mortality in Herbivorous Insects: An Explanation by Modeling the Net Energy Curve"

_toxins, 2024, doi:10.3390/toxins16020072_

Round 1

Reviewer 1 Report

Comments and Suggestions for Authors The authors of the current manuscript offer a theoretical model how the fitness of herbivorous larvae is getting affected by exposure to certain toxins. The manuscript is well-written and fits well the scope of the journal. A number of points of this study---prior to its acceptance---require however a considerable revision. The authors are thus encouraged to address the comments and critique issues listed below.   If many acronyms are in use (in a long review, for instance), the list of all abbreviations as well as of nomenclature is to be provided prior to the bibliography for reader's convenience.   For solid continuous curves of different colors, additionally, different styles or data symbols should be used. Referencing in the text to a curve solely by its color is insufficient.
  Differential equations describing the exponential growth of bacterial colonies at the initial stage in the presence of abundant food sources is similar in essence to the [stochastic differential] equation describing the stochastic process of the so-called geometric Brownian motion (GBM), also giving rise to exponential growth. Interestingly and very relevant to the current material, the GBM under resetting conditions is reminiscent of the model of bacterial growth under abrupt action of toxins or antibiotics killing a large part of the population. In mathematical language this means an abrupt reduction of the value of the process. A number of interesting effects appear in such reset GBM motion, including the phenomenon of breaking of ergodicity. The authors can read more about this in Refs. [DOI: 10.1103/PhysRevE.105.L012106], [https://journals.aps.org/pre/abstract/10.1103/PhysRevE.104.014121], [https://www.sciencedirect.com/science/article/abs/pii/S0960077923008226?via%3Dihub], [​DOI: 10.1103/PhysRevE.106.034137], [https://journals.aps.org/pre/abstract/10.1103/PhysRevE.105.024107] and mention/discuss the analogies with the current model. Such discussion of reset GBM is important to present the current results to a broader community of researchers, that will ultimately increase the interest to the final paper in TOXINS.
  approximative->approximate

Comments on the Quality of English Language

check for typos and suboptimal formulations

Reviewer 2 Report

Comments and Suggestions for Authors

Reviewer Comments

Manuscript Title: How plant toxins cause early larval mortality in herbivorous insects. An explanation by modeling the net-energy curve.

Manuscript ID: toxins-2748138

The manuscript is interesting by delivering mathematical model for relationship between toxin exposure and larval toxicity, yet some minor with grammar errors to be cleared before getting published.

1.      Abstract seems to be so short, possibly add 1-2 lines on the optimal foraging theory employed in this research and the outcome of this research at the end.

2.      Abstract Keywords: Use catchy keywords which are not used in the title of the research, replace larvae or larval mortality with more attractive words relevant to the present study.

3.      The cited reference was not in the Ascending format, Starts from [10,24,62].

4.      Line 16: “In order to” should be “to”.

5.      Line 18: “growth inhibitors” include “,” after “growth inhibitors,”.

6.      Line 19: “leaf chewing” should be “leaf-chewing”.

7.      Line 30: The abbreviation of tripeptide glutathione (GSH) seems to be incorrect please check.

8.      Line 44: Cyclocephala spp. What species they studied? Provide the full species name.

9.       Applying Optimal foraging theory (OFT) to field studies can be challenging. Specially Under Field conditions may introduce complexities that are not considered in theoretical models. Can you address this limitation and their overcome in your conclusion? 

Comments on the Quality of English Language

Minor punctual and grammar corrections are required. 

Reviewer 3 Report

Comments and Suggestions for Authors

Dear Authors,

I read carefully your submitted article Toxins-2748138  and I consider it an interesting contribution  regarding  the  possibility to  create a mathematical  model to "draft" the fitness of herbivorous insect larvae, affected by toxins (ITCs) hydrolyzed by glucosinolates (GLSs) in the host plants. However, by reading the mns, I noticed several points /periods,phrases unclear that needs  to be explained with more details  or re-written. In synthesis, look at the attached word file with my review notes  in the Methos and results chapter (line98, lines 112-114, 166-170).  In the  Discussion paragraph (lines 207-214, 226-227).  Moreover, there are several typing mistakes referring to the latin names of several insect species.

I suggest to consider my notes in order to improve the clearness of the mns and its scientific soundness.

Sincerely

Round 2

Reviewer 3 Report

Comments and Suggestions for Authors

Dear Authors,

I read the new version of the mns Toxin-2748138 after following the notes and suggestions provided by reviewer. I appreciate the efforts done in order to improve the text , considering all periods that needed to be re-write  or to add new lines/phrases also including new proper references. New clarifications have been added in the M&M chapter by recording new references and data referring to the developmental  moth instars studied, clarifications on rearings and growth rates , quality of food for the lepidopteron species considered.  These data have clearly explained and justified the develop of the functions, i.e. energy benefit, toxin cost and foraging cost, that have been considered in this study. In the Discussion section, the Authors  have  pointed out critically ,  how the results obtained can explained the early mortality of larvae due to the plant toxins, through a set of non-autonomous ordinary differential equations (ODEs) according to optimal foraging theory (OFT).  The most important results has been that the net energy intake can become zero at any instar stage, as consequence of the activity and concentration of the toxin. This can produce the immature death of larvae. I am fully satisfied  also for the Author's answers/replies to my questions/notes recorded in the review. I can recognize that new version is well presented and its scientific soundness is clearly  pointed out. The new text can be accepted.

Sincerely
